# Stereotypic Behavior in Fattening Bulls

**DOI:** 10.3390/ani10010040

**Published:** 2019-12-24

**Authors:** Laura Schneider, Nicole Kemper, Birgit Spindler

**Affiliations:** Institute for Animal Hygiene, Animal Welfare and Animal Behavior, University of Veterinary Medicine Hannover, Foundation, D-30173 Hannover, Germany; nicole.kemper@tiho-hannover.de (N.K.); birgit.spindler@tiho-hannover.de (B.S.)

**Keywords:** oral stereotypies, fattening cattle, abnormal behavior, tongue playing, oral manipulation

## Abstract

**Simple Summary:**

Cattle housed under intensive housing conditions may display stereotypic behavior like manipulating objects or body parts of conspecifics with their tongue or rolling and unrolling their tongue repeatedly (so-called tongue playing). These stereotypies may indicate restricted welfare. To our knowledge, there are no studies on the occurrence of stereotypies in fattening cattle. Therefore, this study aimed to analyze the prevalence of stereotypies in 243 fattening bulls housed under different conditions in straw-bedded pens in groups of 14, 16, 22, and 33 animals. The animals in one housing system were fed six times per day, the other animals twice per day. The animals’ behavior was observed at three different stages during the fattening period. Two hundred and thirty-four of 243 bulls were observed performing stereotypies at least once. In the different housing systems, an average of 0.2 to 0.9 stereotypies occurred per animal and hour. The most common stereotypy was manipulating objects, followed by tongue playing and manipulating conspecifics. These results show that stereotypies are a common problem in fattening cattle, occurring frequently under different housing conditions. As this may indicate restrictions in welfare, further studies on stereotypies in fattening cattle are needed in order to detect the reasons for their occurrence.

**Abstract:**

The occurrence of stereotypies in captive animals may indicate restrictions in animal welfare. In cattle, common stereotypies are tongue playing, manipulation of objects, or conspecifics. However, to our knowledge, the occurrence of stereotypies in fattening cattle was only analyzed in studies several decades old. Therefore, this study aimed to analyze the prevalence of stereotypies in fattening bulls housed in different systems. On three German fattening farms, a total of 243 fattening bulls housed in groups of 14, 16, 22, and 33 animals in straw-bedded pens were observed. Behavioral observations were performed via video recordings during three observation periods distributed over the whole fattening period, using a scan sampling technique. In 234 of 243 bulls, stereotypies were observed at least once. During 15.9 ± 2.4% of the scan intervals, stereotypies were observed in at least one animal per pen. Average numbers of stereotypies per animal and hour ranged from 0.2 to 0.9. The most common stereotypy was manipulating objects, followed by tongue playing and manipulating conspecifics. These results indicate that stereotypies are highly prevalent in fattening bulls under current housing conditions. They underline the need for further studies to analyze the causation of stereotypies in order to reduce their frequency.

## 1. Introduction

The development of stereotypies in captive animals has already been described as an indicator of restrictions in animal welfare several decades ago [1,2,3]. According to more recent studies, it is caused by deficits in housing or husbandry, leading to frustration of natural behavior patterns, repeated attempts to deal with problems, and/ or central nervous system dysfunction [4]. Consequently, the occurrence of stereotypies possibly indicates that one of the Farm Animal Welfare Council’s Five Freedoms, namely freedom to express normal behavior, is restricted [5].

According to Broom [2], the welfare of an individual is compromised if it performs stereotypies for more than 10% of its waking life. Wiepkema et al. [3] suggested that stereotypies occurring in more than 1% to 5% of a group of animals could be interpreted as an indicator of compromised animal welfare in a particular housing system.

In cattle, common stereotypies are tongue playing, object licking, and bar biting as well as manipulating different body parts of conspecifics [6,7,8]. Wiepkema et al. [3] defined tongue playing as stereotyped behavior as it is fixed in form and orientation and it is performed repetitively without any obvious function. In contrast, object licking, bar biting, and manipulating body parts of conspecifics are classified as redirected behaviors, being performed due to the absence of adequate substrate and/or occurring very often or intense [3]. In this study, according to Mason [4], redirected behaviors were also interpreted as stereotypies, following the definition of stereotypic behavior as “repetitive behavior induced by frustration, repeated attempts to cope, and/or central nervous system dysfunction”. There are several studies on the occurrence of these stereotypies in calves [9,10,11,12,13,14,15,16], but studies on the occurrence of stereotypies in adult cattle are rare. Nonetheless, available studies agree concerning the high prevalence of stereotypies: In dairy cows, Redbo et al. [17] observed stereotypies in 40 of 95 animals, while Redbo et al. [18] described them in 27 of 37 animals. In a study on heifers, depending on feed, 16 to 44 of 48 animals were observed performing stereotypies [19]. In beef cattle, there are no studies available on the incidence and prevalence of stereotypic behavior according to the EFSA (European Food Safety Authority) Panel on Animal Health and Welfare [20]. Indeed, our literature review only revealed studies published more than 25 years ago [12,13,14,15,16,17,18,19,20,21,22,23]. However, these studies also report a high prevalence of stereotypies with different types of oral stereotypies occurring in all observed bulls.

According to the evaluations of Broom [2] and Wiepkema et al. [3], the literature figures for the prevalence of stereotypies in adult cattle are alarming. As there are no studies on fattening bulls in current housing environments to our knowledge, there is no information available on the current prevalence of stereotypies in fattening cattle. Therefore, this study surveyed the occurrence of stereotypies in fattening bulls housed under different housing conditions.

## 2. Materials and Methods

The study was carried out in accordance with German legislation, the German Animal Welfare Act (German designation: TierSchG) [24], national requirements for animal husbandry (German designation: TierSchNutztV) [25], the Animal Protection Guideline for Fattening Cattle of Lower Saxony, Germany [26], as well as the Council of Europe Convention on the Protection of Animals Kept for Farming Purposes and its recommendations concerning cattle [27]. The study was reviewed and it received approval from the Animal Welfare Officer of the University of Veterinary Medicine Hannover, Foundation (TVO-2017-B5).

The animals observed in the study were 243 Simmental bulls housed on three commercial fattening farms in Germany (Lower Saxony and North Rhine-Westphalia). The bulls were housed in straw-bedded pens in four different housing systems, including groups of 14 (G14), 16 (G16), 22 (G22), and 33 animals (G33). Detailed information on housing, management, and feed composition are listed in Table 1. In G14, the bulls were fed a total mixed ration (TMR) six times a day using an automatic feeding system (AFS; feeding robot Triomatic HP 2 300, Trioliet, Oldenzaal, the Netherlands). In the other housing systems, feed was delivered twice a day using conventional feed-mixer wagons. In addition, feed was pushed up towards the feed barrier several times a day in all systems. The animal/feeding-place ratio was approximately 2:1 in all groups [26]. Water was available ad libitum via two (in G33 four) drinking troughs per pen. Fresh straw was distributed daily with a straw blower. The animals in G33 on farm 3 received additional straw via hayracks.

The animals had arrived at the farms at about six months of age, originating from different farms. They were assigned to groups, these remaining constant until the end of fattening. For the study, four groups of G14 and G22 and two groups of G16 and G33 each were selected. Data acquisition began when the final groups had existed for at least four weeks. It was performed during three observation periods (OP) at an average age of nine months (OP1), 13 to 14 months (OP2), and 17 months (OP3). One of the G33-groups was only observed in OP2 and OP3.

Behavioral observations were performed analyzing video recordings. The animals were videotaped with one video camera per pen (EQ900F, EverFocus Electronics Corporation, Taipei, Taiwan) and an eight-channel hybrid recorder (AXR-108, Monacor International GmbH and Co. KG, Bremen, Germany). The cameras were located above the pens, capturing one full pen each. Individual animals were identified by the color and pattern of their fur. Video analyses were performed using the program Interact (Version 17.0.1.2, Mangold International GmbH, Arnstorf, Germany) for observational research. The activity of the animals was observed for 48 h per OP using a scan sampling technique [28]. At two-minute intervals from 05:30 h to 21:00 h and 10-min intervals during the night (21:00 h to 05:30 h), the number of animals performing stereotypies was recorded. Thereby, it was differed between tongue playing, manipulating objects, and manipulating conspecifics. Tongue playing was defined as a curling and uncurling of the tongue inside or outside the mouth [6]. Manipulating objects referred to object licking (licking, nibbling, sucking, or biting any object, except for feed and straw) [10] as well as bar biting (clamping jaws around a bar and moving head back and forth while chewing on bar) [6]. Manipulating conspecifics was defined as an animal taking a part of the body of a conspecific into its mouth and sucking or biting it [10]. In a second step of video analysis, the behavior of all animals was scanned at individual level at 10-min intervals from 05:30 h to 21:00 h on three consecutive days per OP, using a combination of scan sampling and focal animal sampling [28]. The recorded behavioral patterns were consistent with those of the evaluation at herd level.

The data were analyzed descriptively, using Microsoft Excel (2010). At herd level, the percentage of animals performing stereotypies was averaged per interval. In addition, the percentage of scan intervals between 06:00 h and 23:00 h with stereotypies being observed in at least one animal was calculated per group and OP. Furthermore, the average number of stereotypies per animal and hour was calculated per OP. At the individual level, the percentage of animals observed performing each stereotypic pattern at least once was calculated per group.

## 3. Results

Stereotypies were observed in 234 of 243 bulls (96.3%; Figure 1). In G14 and G16, all animals were found to perform stereotypies, while in G22, all but two, and in G33, all but seven animals were affected. Tongue playing was observed in the greatest percentage of animals, ranging from 81.8% (G33) to 95.5% (G22). Manipulating objects and conspecifics followed with percentages of 42.4% to 92.9% and 34.9% to 71.9%, respectively.

In total, stereotypies were observed in at least one animal during an average percentage of 15.9 ± 2.4% of the scan intervals from 06:00 h to 23:00 h. Differentiating between the four housing systems, these percentages were slightly lower in G14 (14.7 ±.2.6%) and G33 (14.9 ± 2.1%) than in G16 (17.0 ± 1.8%) and G22 (16.9 ± 2.4).

The average number of stereotypies observed per animal and hour was calculated per housing system and OP (Figure 2). Values ranged from 0.2 (G16, OP2) to 0.9 (G16, OP1). The most common stereotypy was manipulation of objects in all housing systems and OP, occurring on average 0.1 to 0.5 times per animal and hour. Tongue playing and manipulation of conspecifics followed, occurring 0.01 to 0.2 times and 0.02 to 0.2 times per animal and hour. Developments over the course of the fattening period varied between the different housing systems. In G22, there was a linear decrease in the average number of stereotypies per animal and hour from OP1 to OP3. In consistence, the average number of stereotypies in G16 was clearly higher in OP1 than in the following OP, but values slightly increased again from OP2 to OP3. In G14, there was no clear development over the course of the fattening period and in G33, the number of stereotypies slightly increased with OP.

The averaged percentage of animals performing stereotypies over the course of the day are displayed in Figure 3. Highest values were observed in all housing systems during the daytime, between 07:00 h and 21:00 h. During the night, the averaged percentage of animals performing stereotypies never exceeded 2% and even periods without any animals performing stereotypies were observed. Maximum values ranged from 4.2% in G33 to 7.8% in G16. In all housing systems with conventional feeding systems delivering feed twice a day (G16, G22, and G33), maximum values occurred during the periods of feed delivery.

## 4. Discussion

This study aimed to survey the prevalence of stereotypies in fattening cattle. It can be concluded that the prevalence is high in fattening bulls housed in various housing systems. Only 9 of 243 bulls were not observed performing stereotypies. All other animals were observed at least once displaying tongue playing or manipulating objects or conspecifics. Consequently, the results of the current study do not indicate a reduced prevalence of stereotypies in fattening cattle in comparison to the studies that were published several decades ago: Sambraus et al. [22] observed 55 fattening bulls aged between 8 and 13.5 months that all displayed tongue playing at least once. Manipulating objects even was observed at least once per day in all bulls, and manipulation of different body parts of conspecifics in up to 49% of the bulls. Graf [21] analyzed the occurrence of manipulating objects and conspecifics in 465 fattening bulls aged between six and nine months and fed with rations with or without hay. He observed the manipulation of objects in 99.1% to 100% of the bulls and the manipulation of conspecifics in 25.2% to 39.4% of the bulls. In the present study, there was no stereotypic behavior that was observed in all animals. However, the highest percentage for any one stereotypic behavior was observed for tongue playing with 81.8% to 95.5% of the animals in the different housing systems, and manipulation of objects also occurred in a considerable percentage of the animals with up to 92.9% of the bulls in G14 exhibiting this behavior. Manipulation of objects in the remaining housing systems as well as manipulation of conspecifics occurred in at least 34.9% and up to 71.9% of the animals. Consequently, the percentage of animals performing stereotypies clearly exceeded 1% to 5%, suggested by Wiepkema et al. [3] as a critical value. Thus, the observed prevalence of stereotypies possibly indicates compromised animal welfare.

According to Broom [2], the occurrence of stereotypies indicates compromised welfare, when individual animals spent more than 10% of their waking life performing stereotypies. Whether this was the case in individual animals in the present study cannot be determined for sure, as the frequency of stereotypies was only analyzed in detail at the herd level. However, stereotypies occurred during an average percentage of up to 17.0 ± 1.8% of the scan intervals from 06:00 h to 23:00 h. This observation, as well as average figures of up to 0.9 stereotypies occurring per animal and hour, indicate that individual animals might have performed stereotypies during more than 10% of their waking life. Furthermore, it has to be taken into account that the data of the present study were acquired using scan sampling. Consequently, the real number of stereotypies is possibly even higher than observed. As stereotypies might be performed only for short periods of time, they are likely to be underestimated by the method of scan sampling due to the low possibility of short behavioral patterns occurring exactly at the sampling point [28].

Consistent with Sambraus et al. [22], Wierenga [23], and Graf [21], manipulation of objects was the most common stereotypy, being observed on average 0.1 to 0.5 times per animal and hour. In the literature as well as in the present study, tongue playing [22,23] as well as manipulation of conspecifics [21,22,23] occurred less often.

Over the course of a day, stereotypies mainly occurred during daylight hours. The low percentage of animals performing stereotypies during the night can be possibly explained by cattle being diurnal feeders with a nocturnal resting period [7,29]. There was a clear connection between periods with a high percentage of animals performing stereotypies and the periods of feed delivery, this being consistent with other studies (calves: [12,13,23], dairy cows: [30,31]). This temporal connection may support the observation of various authors in calves, heifers, and dairy cows that feed influences the occurrence of stereotypies [10,13,14,15,16,18,19,21]. Bergeron et al. [8] lists three hypotheses explaining the connection between feeding behavior and stereotypies in ungulates: Firstly, diets may not fully satisfy them, providing an insufficient gut fill or being deficient in some specific ways. Secondly, diets may take too little time to find, chew, or ruminate in comparison to natural diets, leading to unfulfilled feeding motivations of the animals, independent of gut fill. Thirdly, oral stereotypies may be an attempt to cope with low-fiber, carbohydrate-rich diets that negatively affect gut function. However, the occurrence of oral stereotypies in cattle is likely to be caused by a variety of factors [8,10]. Group size may be one of these. Leruste et al. [10] observed less manipulation of substrates and less tongue playing in calves housed in groups of more than 10 animals. In contrast, Gaude [11] observed more nutritive cross-sucking in calves housed in groups of more than 10 animals. In the present study, no clear influence of group size on the occurrence of oral stereotypies could be observed. There were differences between the different housing systems, but no obvious dependency on group size. Thus, the observed differences are more likely to be caused by other management factors that differed between groups, e.g., farm, space allowance, feeding management, or composition or particle size distribution of the TMR. As the groups observed in this study varied in various aspects, no clear conclusions can be drawn regarding the factors responsible for the development of stereotypies. To determine these, further experimental studies with comparable control groups are required.

The developments over the course of the fattening period also differed between the housing systems. In G14, there were no clear differences between the three OP, and the number of stereotypies per animal and hour stayed at an approximately constant level. In G33, the number of stereotypies slightly increased from OP1 to OP3, being consistent with the observations of Graf (1991) [21]. However, the developments in G16 and G22 were inverse: In G22, the values linearly decreased from OP1 to OP3 and in G16, they were clearly higher in OP1 than in OP2 and OP3. As G16 and G33 were housed on the same farm and received the same feed, neither group size nor feed composition nor management can be the only factor responsible for the development of stereotypies. The only noticeable difference between G16 and G33 was that G33 received additional straw in hayracks. This may explain the lower number of stereotypies occurring in G33 in comparison to G16 in OP1, as more structured feed is likely to reduce the occurrence of stereotypies [10,15,16,19,21]. Nonetheless, it cannot serve as a sound explanation for the values being considerably lower in OP2 and OP3 than in OP1 in G16, nor for the increasing values from OP1 to OP3 in G33. These differences are possibly caused by differing rearing conditions, as the animals originated from different farms. Consistently, Mason [1] stated that stereotypies once developed may be performed later in life in circumstances where the animal’s wellbeing is not at stake. Therefore, the high level of stereotypies observed in this study cannot be interpreted as clear evidence of certain housing systems not being suitable for meeting the animal’s needs. However, they clearly indicate that oral stereotypies are highly prevalent in fattening bulls housed under different conditions. As the occurrence of stereotypies may indicate restrictions in animal welfare occurring at some point in the animal’s life, there is an urgent need to further analyze the causation of stereotypies in order to find ways of reducing their frequency.

## Figures and Tables

**Figure 1 animals-10-00040-f001:**
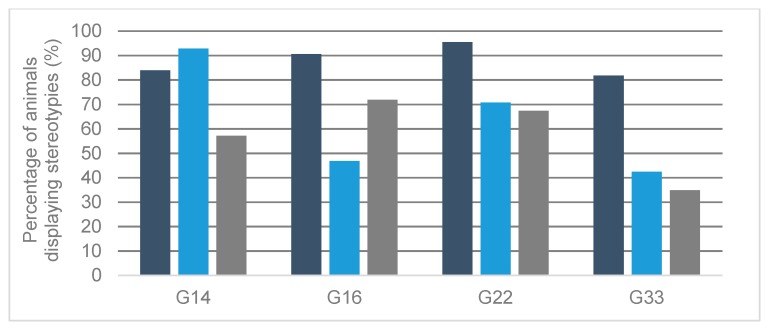
Percentage of animals observed displaying different stereotypic behavioral patterns in different housing systems. G14 = groups of 14 animals, G16 = groups of 16 animals, G22 = groups of 22 animals, G33 = groups of 33 animals. Dark blue = tongue playing, light blue = manipulating objects, gray = manipulating conspecifics.

**Figure 2 animals-10-00040-f002:**
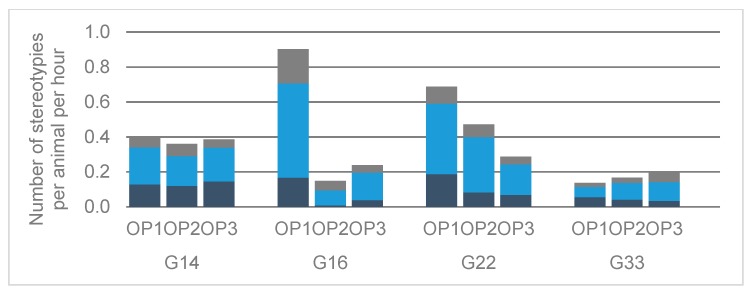
Averaged number of stereotypies observed per animal and hour in different housing systems and observation periods (OP). OP1 = First OP (age of the animals: Eight to nine months), OP2 = second OP (13 to 14 months of age), OP3 = third OP (17 months of age), G14 = groups of 14 animals, G16 = groups of 16 animals, G22 = groups of 22 animals, G33 = groups of 33 animals. Dark blue = tongue playing, light blue = manipulating objects, gray = manipulating conspecifics.

**Figure 3 animals-10-00040-f003:**
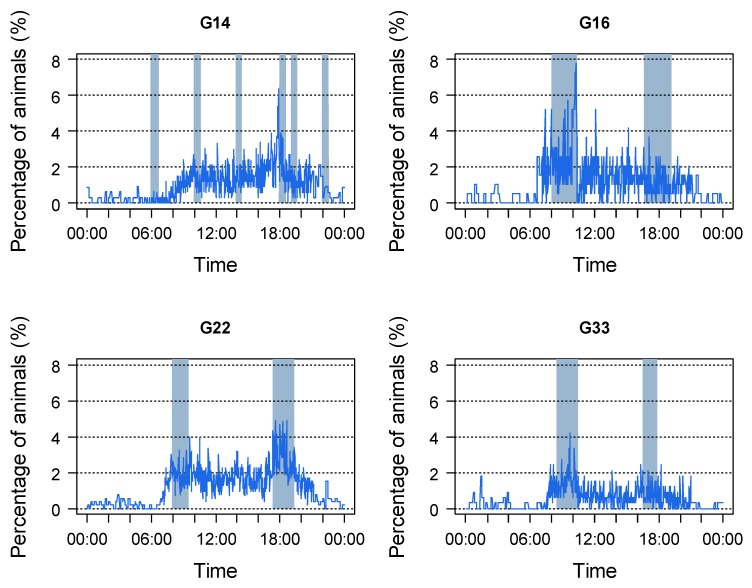
Averaged percentage of animals per pen performing stereotypies over a 24 h period. G14 = groups of 14 animals, G16 = groups of 16 animals, G22 = groups of 22 animals, G33 = groups of 33 animals. Blue areas = periods of feed delivery. Data were averaged for each interval for three observation periods of two days each and two groups of G16 and G33 as well as four groups of G14 and G22 each.

**Table 1 animals-10-00040-t001:** Housing information for the observed animals. TMR = total mixed ration, DM = dry matter.

Farm Information	Farm 1	Farm 2	Farm 3
Number of bulls per group	14	22 ^1^	16, 33
Space allowance per bull (m²)	4	3.5/4.4 ^2^	4.5
Feeding system (number of feed deliveries per day)	Automatic feeding system (6)	conventional feed-mixer wagon (2)	conventional feed-mixer wagon (2)
Groups considered in this study	4	4	2, 2
**Ingredients of the TMR (%)**			
Maize silage	72.5	85.5	84.1
Grass silage	-	0.9	7.7
Barley straw	1.0	-	-
Potatoes	12.7	4.5	-
Concentrated feed/grain	13.2	8.4	7.7
Mineral and vitamin mix	0.6	0.7	0.6
**Chemical composition of the TMR**			
DM (%)	40.5	44.5	35.9
Crude protein (% DM)	12.9	11.6	10.5
Crude ash (% DM)	6.1	6.0	3.8
Crude fat (% DM)	2.3	2.1	3.2
Crude fiber (% DM)	16.1	14.2	16.3
Nitrogen free extractives (% DM)	62.6	66.1	66.2
pH	4.6	4.8	3.9
**Particle size distribution of the TMR (%)**			
Particles remained by 19 mm sieve (long)	10.9	7.9	3.8
Particles retained by 8 mm sieve (medium)	50.5	58.4	57.6
Particles on bottom pen (short)	38.6	33.7	38.6

^1^ One of the G22-groups consisted of 23 animals in observation period (OP) 1. After OP1, one animal was removed from that group. ^2^ The space allowance on farm 2 increased during fattening as the bulls were transferred to larger pens between OP2 and OP3.

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
