# Peer review of "Stereotypic Behavior in Fattening Bulls"

_animals, 2019, doi:10.3390/ani10010040_

Round 1

Reviewer 1 Report

Stereotypical behaviors were observed on 3 farms with 4 different feeding/grouping systems, over the fattening period in bulls. The authors say work hasn't been done in this area for some time and therefore, this is an area of interest to welfare scientists to update information on cattle stereotypies which is lacking.

Though this is valuable to show bulls display stereotypies, this manuscript seems more like an observational assessment of 4 different farm setups, and not an experimental design with control to understand exactly how housing affects stereotypies, even though the objective was to look at sterotypies under different housing conditions. This may be why it was hard for the group to come to any conclusions on housing.

In the abstract write out the 4 different group sizes so different housing conditions are clear.

Material and Methods

Ln 108 though we later know you refer to manipulating objects, conspecifics and tongue rolling as the stereotypical behaviors, you never say these are stereotypies within this section. So just define these are the stereotypies.

Ln 113 this is the only time you mention focal sampling. How many animals/pen were the focal points? Which behaviors were done through this sampling method.

Ln 116 was there no statistical analysis software used to analyze data for statistical differences? No p values are reported throughout.

Discussion

Expand on what specific objects were manipulated in that observation. These were all straw bedded pens. Were bulls seen manipulating/chewing the straw? If so, that could skew the results as straw may be seen as an enrichment and those foraging activities may improve welfare.

Expand on the stereotypies being higher at feeding periods.This was during the actual feeding period not immediately before, correct? Could there be problems with all cattle having access to feed and if some didn't increased stereotypies when 'waiting' for their turn to eat? For example, bar biting seems to increase in sows right before a meal time because they're hungry and anticipating the meal. Some oral stereotypies like tongue rolling therefore, could be associated with feed more than bad welfare.

Ln 212-214 what do you mean here about no difference based on group size? Your housing systems were different group sizes and space allowances, so how is this not your main housing difference besides feed composition? Did references in this section discuss space allowance or just total number of animals in a group, as space allowance has a great influence on stress behaviors.

Ln 227 'structured' what do you mean here? Were G33 bulls not manipulating floor bedding straw more because they received it in a a rack?

Reviewer 2 Report

The problem of abnormal behaviours in domestic cattle has been very important for many years - especially when the animals are kept in intensive housing systems. Therefore the idea of carrying another study in this scientific area, in my opinion, is very desirable.
Unfortunately, the importance of changes that ought to be in the manuscript is too much to accept it to publication in Animals. I mean the following problems:

The Authors not precisely enough defined the key concept of the abnormal animals' behaviours. For example, it is very important to discrimination between stereotypic behaviours (eg tongue rolling) and redirected behaviours (eg cross sucking, intersucking or "manipulating objects"). Because there is generally a problem in definitioning of stereotypies in animals, the Authors should write the definition to which they to refer to. The way of describing cows' abnormal behaviours influences the rest text of the manuscript. It should be improved.

The Authors wrote: "Currently, there are no studies on the occurrence of stereotypies in fattening cattle."
Why such certainty? Can a scientist say like that? This also applies to the rest of the manuscript's text in this context.

There were no actual statistical (non-parametric) methods used to compare animal groups. It was among others necessary because the observations were realized in different environmental conditions. For example, it is very likely that the effect of the barn and observation periods could be statistical significance in this case.

Round 2

Reviewer 1 Report

Thank you for your thorough responses and adding extra clarification to the manuscript. 

Reviewer 2 Report

If the co-reviewers do not submit any important remarks that undermine the value of the current version of the manuscript, I express my positive opinion on its admission to publication in Animals.